# Synthesis and Antibacterial Activity of Cationic Amino Acid-Conjugated Dendrimers Loaded with a Mixture of Two Triterpenoid Acids

**DOI:** 10.3390/polym13040521

**Published:** 2021-02-09

**Authors:** Anna Maria Schito, Gian Carlo Schito, Silvana Alfei

**Affiliations:** 1Department of Surgical Sciences and Integrated Diagnostics (DISC), University of Genoa, Viale Benedetto XV, 6, I-16132 Genova, Italy; amschito@unige.it (A.M.S.); giancarlo.schito@unige.it (G.C.S.); 2Department of Pharmacy (DiFAR), University of Genoa, Viale Cembrano 4, I-16148 Genova, Italy

**Keywords:** new bactericidal agents, antibacterial cationic dendrimers, arginine and lysine-conjugated dendrimers, multidrug-resistant Gram-positive bacteria, *Enterococcus* and *Staphylococcus* genera, commercial ursolic and oleanolic acids

## Abstract

To counteract the growing bacterial resistance, we previously reported the remarkable antimicrobial activity of amino acid-conjugated cationic dendrimers (CDs) against several Gram-negative species, establishing that the cationic lysine was essential for their potency. In this paper, CDs conjugated with lysine and arginine and encapsulating ursolic and oleanolic acids (UOACDs) were assumed to be excellent candidates for developing new antibacterial agents, possibly active against Gram-positive species. Indeed, both the guanidine group of arginine and the two triterpenoid acids are items known for directing antibacterial effects, particularly against Gram-positive bacteria. The cationic dendrimers were obtained by peripheral conjugation with the selected amino acids and by entrapping a physical mixture of the commercial triterpenoid acids. The cationic compounds were characterized and successfully tested against 15 Gram-positive isolates. Interesting minimum inhibitory concentration (MIC) values were obtained for all the dendrimer-drug agents, establishing that the antibacterial activity observed for the UOACDs strongly depended on the density and on the type of the cationic groups of the cationic amino acid-conjugated dendrimers and not on the presence and the release of UOA. Particularly, lysine was critical for potency, while arginine was critical for redirecting activity against Gram-positive species. Especially, a high cationic character, associated with a balanced content of lysine/arginine, produced a remarkable antimicrobial effect (MIC = 0.5–8.7 µM).

## 1. Introduction

The rapid and worldwide increase in antimicrobial resistance among bacterial pathogens, frequently associated with therapeutic failures and high mortality rates, urgently requires alternative curative options able to replace the no-longer-active antibiotics [1].

The most concern regards Gram-negative bacilli, such as *Acinetobacter baumannii, Pseudomonas aeruginosa* and *Stenotrophomonas maltophilia*, which are emerging as clinically relevant superbugs, contributing significantly, with their worrying resistance levels, to numerous therapeutic failures [2]. Today, antibiotic resistance has also become a major problem in the treatment of infections caused by many Gram-positive bacteria. The most important Gram-positive resistant organisms include penicillin-resistant *Streptococcus pneumoniae*, methicillin-resistant *Staphylococcus aureus* (MRSA), *S. epidermidis* (MRSE) and *Enterococcal* species, such as *Enterococcus faecium* and *E. fecalis*, that express high-level resistance to aminoglycosides and/or resistance to vancomycin (VRE) [3].

Frequently, these strains become tolerant to currently available agents, thus requiring innovative therapeutic strategies, including the use of alternative, nonconventional drugs, alone or in combination, and the development of new drugs.

In this regard, natural cationic antimicrobial peptides (CAMPs) [4,5,6,7] represent an appealing class of potent unconventional antimicrobial molecules characterized by a broad spectrum of action, being active on a wide variety of Gram-positive and Gram-negative bacteria, fungi, protozoa, and yeast [8,9]. According to several observations, CAMPs may damage and kill microorganisms by interfering with several specific and crucial bacterial vital processes [10], but mainly, CAMPs are reported as membrane-active compounds and as membrane disruptors [8,9,10,11].

Despite their efficacy and fast action, and the low incidence of the development of resistance [12], CAMPs are endowed with low biocompatibility and high toxicity against eukaryotic cells, which limit their clinical use [8].

In the last twenty years, less toxic, more stable and more low-cost synthetic mimics of CAMPs, including cationic peptides, positively charged polymers, and more recently, positively charged dendrimers, containing different cationic structures, have been developed [8,13]. Concerning dendrimer polymers, they represent, per se, a unique class of macromolecules, very different from linear polymers, endowed with an easily tunable solubility, extensively studied for applications in many sectors, including the biomedical one [14,15,16,17].

Several positively charged dendrimers, including poly(amidoamine) dendrimers (PAMAMs), which displayed very low minimum inhibitory concentration (MIC) values in vitro [18], have been developed and successfully investigated as new unconventional antibacterial agents, to treat infections sustained by multidrug-resistant bacteria [13]. Unfortunately, the low biodegradability, susceptibility to opsonization, high level of hemolytic toxicity, cytotoxicity and fast clearance of non-modified PAMAMs hamper the clinical application of such compounds [14,15,16,17].

Chemical modifications of native PAMAMs were projected to address these issues, but their antibacterial activity was negatively influenced. Although they have still been little studied, polyester-based dendrimer scaffolds, peripherally functionalized with natural amino acids, have been proven to represent very appealing alternative candidates, endowed with considerable antibacterial activity and no cytotoxicity [19,20,21], and with good biodegradability [22,23,24,25].

Against this background, with the aim of developing new biodegradable antibacterial agents active against multidrug-resistant (MDR) Gram-positive species, we prepared three polyester-based-amino-acid-conjugated dendrimers, loaded with a 1:1 physical mixture of commercially available ursolic acid (UA) and oleanolic acid (OA) (UOA) (Figure 1a), and we assessed their antibacterial effects against 15 Gram-positive clinical isolates, with excellent results. Interestingly, marine-derived strains isolated from seawater of the Ligurian west coast were also included since these organisms are frequently found close to coastal areas and may therefore be involved in generating infections in individuals with diminished immune defenses bathing in polluted water. We decided to use a mixture of UO and OA, in place of the pure compounds, because when extracted from medicinal plants, they are frequently obtained as a mixture laborious to separate, and because both of them are endowed with similar beneficial properties, including the antibacterial ones [26].

In particular, the prepared amino acid-conjugated dendrimer-drug systems possessed inner uncharged polyester matrices of the fourth (G4) or fifth (G5) generation (Figure 1b), peripherally esterified with mixtures having different contents of *L*-arginine (R) and *L*-lysine (K) or with R alone. Additionally, they entrapped three, four or eight moles of UOA per mole of dendrimer. They were named G4R(16)K(19)UOA(4), G5R(38)K(30)UOA(8) and G5R(66)UOA(3), the numbers in round brackets being the contents of peripheral residues of amino acids conjugated for each dendrimer and the contents in equivalents of UOA entrapped per dendrimer mole. Their stylized structures have been represented in Figure 2, which also reports the number of moles of UOA loaded by one mole of each dendrimer.

Note that the final dendrimers retained a variable number of free hydroxyls during functionalization, which is indicated with the numbers subscripted close to the OH groups in Figure 2 and which contributed to their water solubility.

## 2. Materials and Methods

### 2.1. Chemistry: Materials and Measurements

All the reagents and solvents were purchased from Merck (Darmstadt, Germany) and used without further purification. Ursolic acid (UA) and oleanolic acid (OA), with purity assessed by gas chromatography analysis, were purchased from Merck, Darmstadt, Germany (commercial cods. U6753 and O5504, respectively). The dialysis bags (D-Tube^TM^ Dialyzer Maxi, MWCO 3.5 kDa) were purchased from Merck (Darmstadt, Germany). The solvents were dried and distilled according to standard procedures. Petroleum ether refers to the fraction with a boiling point of 40–60 °C. The uncharged dendrimer scaffolds G4 and G5 were prepared as previously described, and copies of the NMR spectra are available in the Appendix A [22]. Freeze-drying, centrifugations, HPLC analysis, thin-layer chromatography (TLC), flash chromatography (FC), elemental analyses, melting point analyses, FTIR and NMR spectroscopy of all the dendrimers, dynamic light scattering (DLS) analysis, Zeta potential determinations, and analysis of the in vitro release of UOA from the dendrimer complexes were performed on the instruments and with the same procedures previously reported [22,25,26]. Organic solutions were dried over anhydrous sodium sulfate and evaporated using a rotatory evaporator operating at a reduced pressure of about 1.3–2.7 kPa.

### 2.2. Synthesis of the Amino Acid-Conjugated Cationic Empty Dendrimers G4R(16)K(19), G5R(66) and G5R(38)K(30): General Procedure

A solution of G4 or G5 in dry DMF (25 mg/mL) was added to a 1/1 feed molar ratio mixture of *N*-tert-butyloxycarbonyl (Boc)Arg(NO_2_) and *N*-Boc-Lys or to *N*-BocArg(NO_2_) (1.05 equiv./OH of G4 or G5) and DPTS (0.2 equiv./OH of G4 or G5), and treated with a solution of dicicloexylcarbodiimide (DCC) in CH_2_Cl_2_ (54 mg/mL, 1.21 equiv./OH of G4 or G5). The solution was kept under magnetic stirring at room temperature for 24 h. The precipitated DCU was removed by filtration and washed with fresh acetone (30 mL). The filtrates and washings were combined, concentrated at reduced pressure, taken with the minimum quantity of acetone and filtered through a short silica gel column (h = 8 cm, ϕ = 2 cm) with the help of acetone (25 mL) to remove the last traces of DCU. The solvent was removed by evaporation at reduced pressure to produce solids, which were crushed with Et_2_O, filtered and washed again with fresh Et_2_O to obtained spongy white solids, which were brought to constant weight under vacuum to produce the Boc-protected empty dendrimers, which were subjected to the following deprotection reaction. A solution of Boc-protected dendrimers freshly prepared in 8.8% HCOOH in methanol (35 mg/mL) was added to a slurry of 10% Pd/C (weight of catalyst/weight of substrate = 1/1) in 8.8% HCOOH in methanol (20 mg/mL) and stirred at room temperature overnight. The slurry was then filtered through a silica plug (h = 8 cm, ϕ = 2 cm) to remove the catalyst, which was washed with fresh methanol (30 mL). The filtrate and washings were combined and evaporated at reduced pressure to remove the solvent. The crude product was dissolved in dry methanol (6 mL) and treated under stirring at room temperature for 24 h with an excess of acetyl chloride (350 µL). After the removal of the solvent at reduced pressure, the solid residue was washed in excess acetone under stirring for 3 h, separated from the solvent by decantation and brought to constant weight at reduced pressure. The unprotected dendrimers were obtained as hydrochloride salts and were further purified by dissolution in water (5–10 mL), centrifugation (3400 rpm, 15 min) to remove impurities, and lyophilization. They were stored in a dryer on P_2_O_5_.

Dendrimer G4[Arg(16)Lys(19)OH(13)]. Very hygroscopic fluffy solid, 69% isolated yield. FTIR (KBr, cm^-1^): 3431 (NH_3_^+^), 1744 (C=O ester), 1628 (NH). ^1^H NMR (DMSO-*d6*, 300 MHz): δ 0.90–2.07 [m, 316H (CH_3_ of dendrimer + CH_2_CH_2_ Arg + CH_2_CH_2_CH_2_ Lys)], 2.76 (m, 38H, CH_2_NH^3+^ Lys), 3.21 (m, 32H, CH_2_NH Arg), 3.54–3.82 (m, 26H, CH_2_OH), 3.80–4.80 [m, 208H (CH_2_O of dendrimer + CHNH_3_^+^ Lys + CHNH_3_^+^ Arg + 13OH)], 7.60–9.20 [m, 242H (^δ^NH + ^ω’^NH_2_^+^ + ^ω^NH_2_ + ^α^NH_3_^+^ Arg and ^α^NH_3_^+^ + ^ε^NH_3_^+^ Lys)].

Dendrimer G5[Arg(66)OH(30)]. Very hygroscopic fluffy solid, 46% isolated yield. FTIR (KBr, cm^-1^): 3402 (NH_3_^+^), 1750 (C=O ester), 1652 (NH). ^1^H NMR (DMSO-*d6*, 300 MHz): δ 0.79–1.39 (m, 282H, CH_3_ of dendrimer), 1.47–2.00 (m, 264H, CH_2_CH_2_ Arg), 3.21 (m, 132H, CH_2_NH Arg), 3.47 (br, 60H, CH_2_OH), 3.80–4.71 [m, 414H (CH_2_O of dendrimer + CHNH_3_^+^ Arg + 30 OH)], 7.81 and 9.55 [m, 528H (^δ^NH + ^ω’^NH_2_^+^ + ^ω^NH_2_ + ^α^NH3^+^ Arg)].

Dendrimer G5[Arg(38)Lys(30)OH(28)]. Very hygroscopic pale yellow fluffy solid, 51% isolated yield. FTIR (KBr, cm^-1^): 3411 (NH_3_^+^), 1743 (C=O ester), 1631 (NH) ^1^H NMR (DMSO-*d6*, 300 MHz): δ 0.90–2.05 [m, 614H (CH_3_ of dendrimer + CH_2_CH_2_Arg + CH_2_CH_2_CH_2_ Lys)], 2.75 (m, 60H, CH_2_NH_3_^+^ Lys), 3.18 (m, 76H, CH_2_NH Arg), 3.39–3.55 (m, 56H CH_2_OH), 3.80–4.70 [m, 418 H (CH_2_O of dendrimer + CHNH_3_^+^ Lys + CHNH_3_^+^ Arg + 28 OH)], 7.60–9.20 [m, 484H (^δ^NH + ^ω’^NH_2_^+^ + ^ω^NH_2_ + ^α^NH_3_^+^ Arg and ^α^NH_3_^+^ + ^ε^NH_3_^+^ Lys)].

### 2.3. Preparation of the Physical Mixture of UA and OA (1:1) (UOA)

Exactly weighed amounts of UO (500 mg, 1.09 mmol) and of OA (500 mg, 1.09 mmol) were transferred into a mortar. Then, by using the pestle of the mortar, the two solids were carefully mixed, obtaining a fine dispersion, which was furtherly homogenized in a mixer. The mixture obtained was characterized by NMR analysis, which confirmed the 1:1 ratio of the two constituents.

### 2.4. Preparation of UOA-Loaded Dendrimers (UOACDs): General Procedure

A solution of dendrimers in dry MeOH (2.0 mg/mL) was added to the UOA 1:1 physical mixture (9 equiv.) of commercial UA and OA. The solution was kept under vigorous magnetic stirring at room temperature for 72 h in the dark. Then, after the removal of the solvent at reduced pressure, the obtained white solids were suspended in dichloromethane (DCM) overnight to wash away the free UOA not entrapped. The solids not dissolved were decanted; the DCM was separated and evaporated to obtain a white solid identified as UOA by IR analysis.

The residual solids were brought to constant weight and then dissolved in MeOH/H_2_O and precipitated in acetone in a centrifuge tube. After two cycles of centrifugation (3400 r/min) and washings with acetone, the wet solids were brought to constant weight under reduced pressure and then stored on P_2_O_5_ in a dryer. The physicochemical properties and spectral data of the obatained G4 and G5 UOACDs are reported in Table 1.

### 2.5. Microorganisms

A total of 17 isolates, comprising two strains of clinical Gram-negative bacteria and 15 isolates belonging to 4 Gram-positive species, were used in this study. The gram-positive bacteria included 12 strains of clinical origin and 3 of marine origin, isolated from seawater of the Ligurian west coast. All the isolates were identified with a VITEK^®^ 2 (Biomerieux, Firenze, Italy) or the matrix-assisted laser desorption/ionization time-of-flight (MALDI-TOF) mass spectrometric technique (Biomerieux, Firenze, Italy). Of the tested organisms, 2 strains were of the *Enterobacteriaceae* family (one *E. coli* and one *K. pneumoniae* resistant (KCP)); 6 strains belonged to the *Staphilococcus* genus, including 2 methicillin-resistant *S. auresus* (MRSA) and one susceptible, and 2 methicillin-resistant *S. epidermidis* (MRSE) and one susceptible; and 9 strains were of the *Enterococcus* genus (4 *E. faecalis* resistant to vancomycin (VRE), including strains 19 and 51 of marine origin, and one susceptible, and 3 *E. faecium* VRE, including strain 3 of marine origin, and one susceptible).

### 2.6. Antimicrobial Assays

The minimal inhibitory concentrations (MICs) of the three UOACDs, UOA and three empty dendrimers on the pathogens were determined following the microdilution procedure detailed by the European Committee on Antimicrobial Susceptibility Testing EUCAST [27].

Briefly, overnight cultures of bacteria were diluted to yield standardized inocula of 1.5 × 10^8^ colony-forming units (CFU)/mL. Aliquots of each suspension were added to 96-well microplates containing the same volumes of serial 2-fold dilutions (ranging from 1 to 1024 µg/mL) of each dendrimer to yield a final concentration of about 5 × 10^5^ cells/mL. The plates were then incubated at 37 °C. After 24 h of incubation at 37 °C, the lowest concentration of dendrimer that prevented visible growth was recorded as the MIC. Concerning these early investigations, all the MICs were obtained in triplicate, the degree of concordance was 3/3 in all the experiments, and the standard deviation (±SD) was zero.

### 2.7. Statistical Analysis

Data are expressed as means ±SDs. Concerning the MIC values, the experiments were performed in triplicate, the concordance degree was 3/3, and the ±SD was zero.

## 3. Results and Discussion

### 3.1. Positively Charged Amino Acid-Conjugated Dendrimer-Drug Compounds Designed for This Study

Concerning the idea of encapsulating UOA, it was thought that the presence of moieties of these triterpenoid acids, reported to be active against species belonging to the genera *Staphylococcus* and *Enterococcus* [28,29], could be helpful in achieving our goal. Moreover, note that by encapsulating the UOA mixture in the cationic dendrimers, we were successful in solubilizing the water-insoluble UA and OA, otherwise not bioavailable and, even if active in vitro, not administrable in vivo. Interestingly, in view of a future possible clinical use of these materials, their water solubility will instead allow an easy in vivo administration.

Regarding the design of the chemical structures of the dendrimer scaffolds, we made the following considerations. In the sector of antibacterial cationic dendrimers, and particularly of those modified with amino acids, high generations translate into a high multivalence, a high content of amino acids and, therefore, a high density of cationic charge [9,18,21,30]. A high cationic charge is an essential requirement for the interaction of the compounds with bacterial membranes and for them exerting their harmful action on these structures vital for pathogenic microorganisms, simply on contact. Consequently, two out of the three dendrimers were prepared to be of the fifth generation (G5), using, as functionalizing molecules, amino acids with two amino groups; in our case, this would have meant a high number of cationic groups. In addition, since in our previous study, G5 dendrimers containing lysine and histidine turned out to be highly selective for non-fermenting Gram-negative species, firstly, we decided to change the amino acid that proved to contribute little to the antibacterial activity (histidine) with arginine.

Secondly, we decided to also prepare a lower generation (G4) dendrimer, to explore whether a significantly reduced content of cationic groups and a less sharp cationic character could allow for the maintenance of significant antibacterial activity and could help to redirect it towards pathogens with a lower negative charge on the external layers, as happens in Gram-positive species. Since lysine was previously found to be essential for good antibacterial activity [21], we decided to maintain it as a functionalizing molecule in the dendrimers of the fourth and in one of the fifth generation and to associate lysine with arginine. Since the guanidine group present in the structure of arginine, at physiologic pH, provides guanidinium groups, which are reported to be more effective against Gram-positive than against Gram-negative bacteria [8,9], the presence of arginine in place of histidine would have helped to obtain antimicrobial agents active against Gram-positive bacterial strains. Finally, to evaluate the intrinsic efficiency of arginine, the second G5 dendrimer was functionalized exclusively with arginine.

Finally, we highlight that the three UOACDs of this paper were obtained without using UOA extracted from plants, but using a physical mixture of UO and OA purchased from Merck. The mixture was artificially prepared with the aim of reproducing the one commonly provided by the nature, in which UO and OA are in the ratio of 1:1.

### 3.2. Positively Charged Amino Acid-Conjugated Empty Dendrimers

For the preparation of the fourth- and fifth-generation dendrimers containing arginine and lysine residues, the grafting of amino acids was successful with DCC and DPTS as the activator and catalyst, respectively, instead of EDC and DMAP as frequently suggested. The analysis of the ^1^H NMR spectra of both the Boc-protected intermediates and final cationic dendrimers evidenced that not all the OH groups at the periphery were reactive towards the protected amino acids, and a number of free hydroxyl groups were detected in all the compounds. By the integration and comparison of opportunely selected proton signals, it was possible to obtain the number of residual OH groups and the contents of amino acids of the Boc-protected forms of the three dendrimers. The deprotection of the Boc-dendrimer intermediates was performed in two consecutive steps by removing, first, the NO_2_ group with formic acid in the presence of Pd/C, followed by treatment with CH_3_COCl and methanol to remove the Boc groups. The composition of amino acids and the number of residual OHs were obtained from the integrals of opportunely selected proton signals in the ^1^H NMR spectra.

The results confirmed the previous data obtained for the Boc-protected compounds, and were employed to calculate the molecular weights (MW) of the dendrimers and the reaction yields. As an example, in the SI, an image is reported, which shows the spectral modifications that occurred both in the phase of esterification and in the phase of protecting-group removal, during the preparation of the cationic empty dendrimer G5R(38)K(30)OH(28), indicating the signals whose integrals were used to calculate its peripheral composition (Appendix A). A similar scenario was also observed for the other two dendrimers.

The polycationic dendrimers (Appendix A) were obtained as hygroscopic glassy solids after freeze-drying and were soluble in MeOH, DMSO, DMF, and H_2_O and insoluble in toluene, Et_2_O, THF, dioxane, CH_2_Cl_2_, CHCl_3_, EtOAc, acetone, acetonitrile and EtOH. In addition, in order to have additional evidence of the prepared structures and of the composition at the periphery, the MWs of the cationic dendrimers were obtained experimentally, by the titration of amine hydrochlorides with HClO_4_ solutions in AcOH in the presence of mercuric acetate and quinaldine red as an indicator [22,25]. The method is simple and low cost, and its accuracy was secured by a sharp endpoint of titration, while its reliability has been demonstrated by the reproducibility of its results [22,25].

The particle hydrodynamic size (diameter) and zeta potential of the prepared dendrimers were determined by dynamic light scattering (DLS) analysis. As expected, the mean diameter of the fourth-generation sample was lower than that of the fifth-generation ones, and also depended on the type of amino acid present on the periphery. They were 4.6, 5.1 and 5.3 nm, respectively, for G4R(16)K(19), G5R(66) and G5R(38)K(30), and comparable to the diameters reported for NH_2_ dendrimers of the same generation [22]. The surface charge was positive (+34.8, +50.0 and +51.8 mV) and in perfect agreement with data reported in the literature for G4 and G5-PAMAM-NH_2_ dendrimers. They showed higher values for the G5 samples than G4, increased with an increase in N (the number of cationic groups) and, being higher than the critical value of 30 mV, should assure a good stability in solution.

### 3.3. Physical Mixture of Commercial UA and OA, 1:1 (UOA)

The desired mixture, 1:1, was obtained by homogenizing, both manually and mechanically, UA and OA, and was analyzed by the NMR technique to establish its actual ratio. The ^1^H NMR spectrum, recorded in CD_3_OD/DMSO-*d6,* confirmed the 1:1 ratio.

### 3.4. Preparation of Dendrimers Loaded with the Physical Mixture of UA and OA, 1:1 (UOA)

The UOACDs prepared in this study are original antibacterial cationic agents, because in place of using commercially available highly cationic and non-biodegradable PAMAM scaffolds, as host macromolecules for the transport and delivery of UOA, they were totally synthesized in our laboratory and harmonized hydrolysable ester-based uncharged matrices, with cationic shells made of natural amino acids. Note that, in view of a clinical use of the prepared compounds, the biodegradability of the uncharged dendrimer architectures should assure a minor risk of permanent damage to the cellular membranes of mammalian cells and minor toxicity.

Robust and optimized synthetic procedures were performed to prepare the UOA-loaded polyester-based amino acid-conjugated cationic dendrimers as described in detail in Section 2.

Concerning the complexation reaction, it can be rationally assumed that the UOA mixture could be entrapped due to the presence of the several amino groups, capable of establishing hydrogen bonds and/or electrostatically interacting with the weak triterpenoid acids. In this regard, the literature reports that in dendrimer/drug systems [31,32,33], such as cationic PAMAM/weakly acidic drug moieties (ibuprofen, DOX, and MTX) [34,35,36], noncovalent forces, such as hydrogen bonding and ionic interactions, are among those mechanisms suggested to play a decisive part in dendrimer/drug complexation. The certainty that UOA was actually adsorbed/encapsulated in the cationic dendrimers and not simply physically mixed was ensured by the purification procedures for the UOACDs. In this regard, as described in the experimental part (Section 2.4), the crude solids obtained at the end of the encapsulation reaction were washed with DCM overnight to eliminate the still-free UOA fraction, which was then recovered by the washings and identified by FTIR.

As for the physicochemical characterization of G4R(16)K(19)UOA(4), G5R(38)K(30)UOA(8) and G5R(66)UOA(3), several techniques, including FTIR and NMR analysis, were employed to confirm the structures and to establish the number of UOA molecules that were entrapped in the dendrimer’s structure. In addition, the MW values, particle sizes and Z-potentials, as well as the profiles of the release of UOA from the hosting dendrimers, were determined.

Table 2 reports the main structural characteristics of the prepared UOACDs and their physicochemical properties, which were determined on the same instruments and following the same procedures previously described [26].

In particular, FTIR was not significantly diagnostic enough to confirm the formation of the complexes, but ^1^H NMR spectroscopy was useful for having both qualitative and quantitative information about the composition of the prepared compounds.

As an example, in Figure 3, the ^1^H NMR spectrum of G4R(16)K(19)UOA(4) is shown. The spectra of the other two UOACDs are available in the Appendix A.

The peaks under 1.0 ppm, not present in the parent dendrimer spectrum (Appendix A) and belonging to the UOA mixture, correspond to the seven CH_3_ groups and H (C(5)) of UA and UO, for a total of 22 H [37], while the broad peak at 3.7 ppm and not present in the UOA spectrum (Appendix A) corresponds to the methylene in the CH_2_OH groups of the dendrimer. The peaks under 1.00 ppm and the broad peak around 3.7 ppm were very diagnostic for confirming the success of the encapsulation reaction, and the integral values associated with these peaks were used to estimate the numbers of UOA units encapsulated per dendrimer mole of all the compounds. Briefly, the numbers of UOA units per complex mole were obtained by comparing the integral values of the CH_2_OH groups of the dendrimer scaffolds at 3.6–3.8 ppm, which were associated with 13, 60 and 56 proton atoms for G4R(16)K(19)UOA(4), G5R(66)UOA(3) and G5R(38)K(30)UOA(8), respectively, and the values of the integrals of the peaks between 0.72 and 0.98 ppm, which were associated with 22 H.

As expected, among the dendrimers containing both *l*-lysine and *l*-arginine, the one of the fifth generation showed a better efficiency in complexing UOA, due to its larger number of cationic groups, which dictated a greater possibility for electrostatic interactions and the formation of hydrogen bonds.

Surprisingly, the dendrimer containing only *l*-arginine, although of the fifth generation and with a high number of cationic groups, showed very poor ability for trapping UOA, which was lower than that of G4R(16)K(19)UOA(4), which only had 70 cationic groups. These findings established that not only the number of cationic groups but also the type of cationic groups could influence the encapsulation ability. The estimated UOA units loaded per dendrimer mole were useful for estimating the MWs of the prepared UOACDs (Table 2). In particular, after having estimated the UOA units loaded per dendrimer mole, it was possible to make an estimate of the MWs of the compounds by simply summing the MWs of the forerunner dendrimers with the MWs of the UOAs multiplied by the numbers of complexed units as deduced from the NMR spectra. Since agents aimed at future clinical applications have to possess peculiar requisites, in terms of water solubility, particle dimensions and surface charges, the determination of their values and a brief discussion of the results obtained is herein mandatory. The particle sizes of the UOACDs ranged from 16 to 25 nm, while the Z-potentials were positive and ranged from 25 to 34 mV. It is known that small particles assure minor tissue toxicity, but extremely minute particles could easily undergo hepatobiliary and renal clearance [36]. A correct balance to minimize both tissue toxicity and fast clearance by the mononuclear phagocytic system (MPS) is the best solution, and, as reported, particles less than 100 nm and greater than 20 nm in size could be a good choice [38,39].

The average particle size of the UOACDs was, in any case, less than 100 nm, thus assuring that during a possible in vivo administration, they would not generate administration embolism, thus also being suitable for intravenous or intraperitoneal administration to the patient. Unfortunately, the G5R(66)UOA(3) particles displayed a mean size < 20 nm, and the short circulation time could affect their efficiency in vivo. The surface charge of the UOACDs had higher values for the G5 samples and increased with an increase in the number of cationic groups. The UOACDs displayed high water solubility, providing solutions stable over time, as shown by Z-potential values around 30 mV.

Since to explicate their pharmacological effects, entrapped drugs need to be released from the reservoir, the in vitro profile of UOA release from the prepared compounds was studied in buffer solutions at pH = 7.4, adding to the incubation medium ethanol, 20% *v*/*v*, to favor its solubility. Aliquots were taken out of the medium at fixed time points (1, 2, 5, 10, 24 and 48 h), the released triterpenoid acids were quantified by HPLC, and the results are reported as micrograms of UOA released from 10 mg of UOACD as a function of time in Appendix A [26]. As the data shown in Appendix A highlight, all the profiles of the drugs released from the cationic dendrimers showed a three-phase pattern. The first phase was resolved after the first hour of incubation at pH = 7.4, when the release was almost null. Then, a very fast release phase started, while after five hours, a sustained release phase took place. The amount of UOA released from the G4-UOA dendrimer was more than that released from the G5-UOA ones, and the release profile showed a dependence on the N value, due to the stronger interactions of the free UOA mixture with dendrimers having more cationic groups.

### 3.5. Antimicrobial Activities of G4R(16)K(19), G5R(38)K(30) and G5R(66)

The MIC values for all the three dendrimers were obtained by analyzing a total of 17 strains including 14 isolates of clinical derivation and 3 of marine origin. From the preliminary results obtained for *E. coli* and *K. pneumoniae*, selected as representatives of Gram-negative bacteria, it was established that CDs loaded with UOA were not effective in inhibiting their growth. For this reason, their activity was studied in more depth against Gram-positive species, towards which significant inhibitory properties have been reported, especially against 15 strains, representatives of the genera *Enterococcus* and *Staphylococcus* (Table 3). In this regard, it was considered essential to establish whether the antibacterial activity exerted by the dendrimer complexes was due to a synergistic activity of the UOA with the cationic dendrimers, to only the cationic dendrimers or to the UOA released. Rational speculations were possible by simply comparing the MIC values of the free UOA, the MIC values displayed by the UOACDs and the maximum concentrations of UOA obtainable from these antibacterial agents based on the observed MIC values (Table 3). The maximum concentrations of UOA obtainable were calculated considering the UOA released after 24 h from 10 mg of each UOACD (Table 2). Interestingly, it can be asserted that the cationic amino acid-conjugated dendrimers themselves possessed antibacterial activity, that the presence of UOA did not contribute to the observed antibacterial potency, and that the inhibitory activity reported for the UOACDs was solely due to the cationic structures. Indeed, the data obtained showed that, with the exception of one isolated case, the maximum concentration of UOA obtainable on the basis of the MIC values reported (Table 3) was far below the concentrations of free UOA (MIC values of free UOA, Table 3) necessary to inhibit bacteria. For having an experimental confirmation of this speculation, the antibacterial activities of the cationic empty dendrimers were also evaluated against the same bacterial strains in the same conditions, and the results are reported in Table 4. While the MIC values observed were identical to those displayed by the correspondent UOACDs when expressed in µg/mL, when expressed in µM, even if no substantial differences were observed, the MIC values of the empty dendrimers were slightly higher, due to the low MWs of the samples. However, these findings did not establish that dendrimers could also be active at lower micromolar concentrations, which we could not observe, due to the serial 2-fold dilution method adopted to measure the MICs. Evidence supporting this hypotesis was obtained by administering to all the strains the micromolar concentrations observed for the MICs of the UOACDs, and a mainained inhibition of the bacterial growth was detected, thus confirming that the antibacterial activity displayed by the complexes was attributable to the cationic dendrimers only. Furthermore, by a comparison of the overstimated MIC values (µM) displayed by the empty dendrimers with those of commercial antibiotics, the MICs of the G4 and G5 CDs against the MDR bacteria of all the species turned out to be far lower than those of the considered antibiotics, thus asserting the high potency of our antibacterial agents against resistant species.

In particular, G5R(66)UOA(3), containing only arginine, showed a similar potency against *Enterococci* and *Staphylococci*, exhibiting comparable MIC values, with a very narrow distribution, towards both the genera. Its activity against *Staphylococci* was lower than that of G5R(38)K(30)UOA(8), but with the exception of *S. epidermidis* 119, for which it was higher than that of G4R(16)K(19)UOA(4), probably due to the lower density of the cationic charge of the last dendrimer (70 cationic groups versus 132). On the contrary, against all the *Enterococci*, G5R(66)UOA(3) showed much lower antibacterial potency than both the G4RK dendrimer (less cationic) and the G5RK dendrimer (with a similar cationic charge). From these findings, it can be concluded that the genus *Staphylococcus* was more susceptible to the density of the cationic groups present in the dendrimer structure, while the genus *Streptococcus* was more susceptible to the presence of *L*-lysine.

The greatest antimicrobial activity of all the antibacterial agents was observed against the *S. epidermidis* 127 (MRSE) strain, which was the most susceptible isolate of the genus *Staphylococcus* to all the UOACDs, with an order of potency of G5R(38)K(30)UOA(8) > G5R(66)UOA(3) > G4R(16)K(19)UOA(4) (Table 3). G4R(16)K(19)UOA(4) and G5R(38)K(30)UOA(8), all of these containing both lysine and arginine, and being much more active against the *Enterococcus* genus than that of *Staphylococcus*, similarly to the free UOA mixture, displaying very low MIC values (2.2–4.4 and 0.5–1.1 µM, respectively). The most effective compound was G5R(38)K(30)UOA(8), which also showed low MIC values against the *Staphylococcus* genus (2.2–8.7 µM).

It is noteworthy that the antibacterial activity of these cationic dendrimers was evident against susceptible genera, regardless of the patterns of resistance to current antibiotics of the isolates evaluated. Moreover, it can be stated that the antimicrobial activity of the herein-developed cationic materials depended mainly on the density of their cationic charge (and therefore on the number of cationic groups) and on the type of peripheral amino acids (i.e., on the type of ammonium group responsible for the cationic character). From the MIC values reported in Table 3, it appears that, against the genus *Staphylococcus*, the antibacterial activity of the UOACDs was proportional to the number of their cationic groups, while against the genus *Enterococcus*, good antibacterial activity was due to the presence of the cationic groups of lysine. Indeed, G5R(66)UOA(3), not containing lysine, although possessing a number of cationic groups comparable to that of G5R(38)K(39)UOA(8), showed much lower antibacterial activity against all the isolates of the genus *Enterococcus* (MIC values up to 37 times lower). Moreover, G4R(16)K(19)UOA(4), despite possessing a far lower number of cationic groups than G5R(66)UOA(3), was significantly more active against all the *Enterococcus* strains analyzed, probably due to the presence of lysine. Finally, since these first results show that all the dendrimer complexes were ineffective against the Gram-negative strains considered, it can be established that the presence of the arginine guanidine group contributed to making the GCDs selective, in particular, for the Gram-positive species.

In conclusion, we have identified new and unconventional antimicrobial agents that, being specifically active against the main and alarming species of Gram-positive bacteria, could replace traditional antibiotics usually employed against *Staphylococci* and *Enterococci*, including oxacillin and vancomycin, today ineffective against such multiresistant strains.

More in-depth investigations, such as time-killing experiments and turbidimetric studies, are underway to better characterize the activity of the developed CDs and to unveil their mechanisms of action at the molecular level.

## Figures and Tables

**Figure 1 polymers-13-00521-f001:**
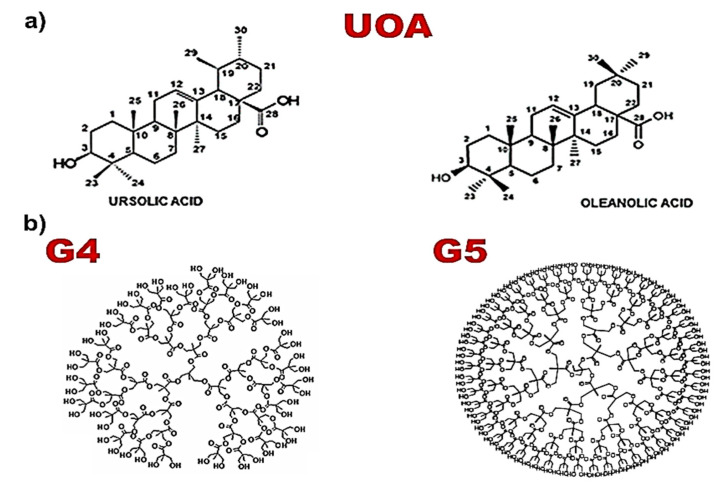
Structure of commercially available ursolic acid (UA) and oleanolic acid (OA), loaded as 1:1 mixture into the cationic dendrimers (CDs) (**a**); structure of the uncharged fourth (G4)- and fifth (G5)-generation polyester-based inner scaffolds of the three CDs (**b**).

**Figure 2 polymers-13-00521-f002:**
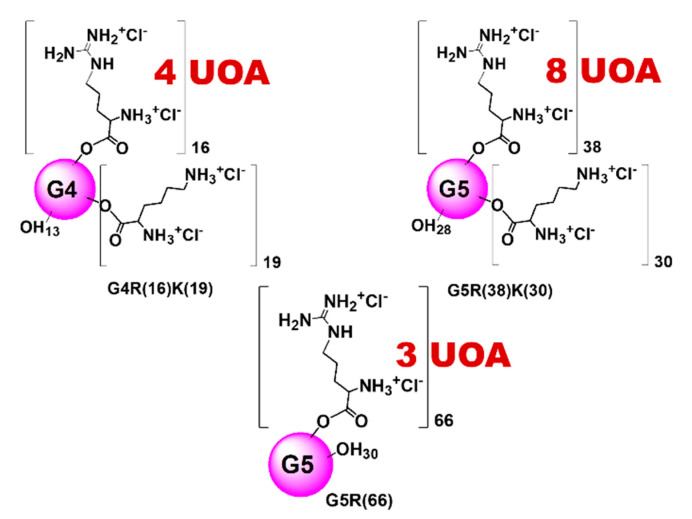
Schematic representations of the three CDs loaded with UA and OA (UOA) ((G4R(16)K(19)UOA(4), G5R(38)K(30)UOA(8) and G5R(66)UOA(3)) prepared in this study.

**Figure 3 polymers-13-00521-f003:**
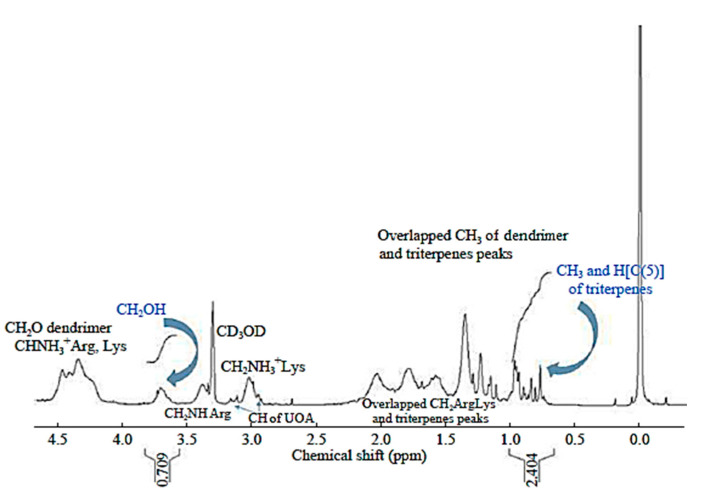
^1^H NMR spectrum of G4R(16)K(19)UOA(4) (300 MHz, CD_3_OD).

**Table 1 polymers-13-00521-t001:** Physicochemical properties and spectral data of the obatained G4 and G5 CDs conjugated with lysine, arginine and encapsulating ursolic and oleanolic acids (UOACDs).

Compound	mg, mmol Yield %	Physical State	FTIR (KBr, cm^−1^)	^1^H NMR (300 MHz, CD_3_OD, *δ*, ppm)
G4[R(16)K(19)OH(13)UOA(4)]	27.0, 0.0018 90	Slightly hygroscopic off-white glassy solid	3431 (OH and NH_3_^+^) 1747 (C=O ester) 1631 (NH)	0.78–0.99 (several s, 88H, H of triterpenoids) 1.00–2.40 (m, 404H, H of dendrimer + H of Arg + H of Lys + H of triterpenoids) 2.94–3.17 (m, 46H, H of Lys + H of triterpenoids) 3.30–3.50 (m, 32H, H of Arg) 3.54–3.82 (m, 26H, H of dendrimer) 4.10–4.50 (m, 195H, H of dendrimer + H of Lys + H of Arg) 4.59–4.71 (dd, 4H, H of triterpenoids) 5.22 (q, 4H, H of triterpenoids)
G5[R(66)OH(30)UOA(3)]	44.2, 0.0016 100	Slightly hygroscopic off-white fluffy solid	3392 (OH and NH_3_^+^) 1743 (C=O ester) 1663 (NH)	0.78–0.97 (several s, 66H, H of triterpenoids) 1.00–2.30 (m, 612H, H of dendrimer + H of Arg + H of triterpenoids) 3.27 (m, 132H, H of Arg) 3.68–3.74 (br, 60H, H of dendrimer) 4.04–4.63 (m, 384H, H of dendrimer + H of Arg) 5.22 (q, 3H, H of triterpenoids)
G5[R(38)K(30)OH(28)UOA(8)]	20.5, 0.00070 64	Slightly hygroscopic pale yellow fluffy solid	3431 (OH and NH_3_^+^) 1747 (C=O ester) 1635 (NH)	0.75–0.98 (several s, 176 H, H of triterpenoids) 1.00–2.40 (m, 790 H, H of dendrimer + H of Arg + H of Lys + H of triterpenoids) 2.95–3.16 (m, 76H, H of Lys + H of triterpenoids) 3.38 (m, 76H, H of Arg) 3.68 (m, 56H, H of dendrimer) 4.10–4.50 (m, 390 H, H of dendrimer + H of Lys + H of Arg) 4.58–4.70 (dd, 8H, H of triterpenoids) 5.22 (q, 8H, H of triterpenoids)

**Table 2 polymers-13-00521-t002:** Main structural characteristics and physicochemical properties of UOA-loaded cationic dendrimers prepared in this study.

Features	G4R(16)K(19)UOA(4)	G5R(38)K(30)UOA(8)	G5R(66)UOA(3)
Arginine, lysine, residual hydroxyl units	16, 19, 13	38, 30, 28	66, 0, 30
UOA moles per dendrimer mole	4	8	3
UOA loading % (*wt*/*wt*)	12.6	12.7	5.0
Dendrimer surface covered (%)	63	69	71
Cationic groups	70	136	132
Molecular weight	14,600	29,300	27,400
Z-potential (mV)	24.8	31.8	34.0
Z-Ave size (nm)	24.9	20.3	16.1
UOA released by complexes after 24 h (µg/10 mg)	75.5	65.9	65.2

**Table 3 polymers-13-00521-t003:** Minimum inhibitory concentration (MIC) values of the three prepared UOACDs and of UOA, expressed as µg/mL and millimolarity (mM), against the Gram-positive strains tested in the study, and the maximum concentrations of UOA released according to the MIC values observed for each dendrimer complex and according to its release profile as reported in Table 2.

	UOA MW 456.7	G5R(66)UOA(3) MW 27,400	Max UOA Released ^1^	G4R(16)K(19)UOA(4) MW 14,600	Max UOA Released ^2^	G5R(38)K(30)UOA(8) MW 29,300	Max UOA Released ^3^
**MIC values ^4^**	µg/mL, µM	µg/mL, µM	µg/mL, µM	µg/mL, µM	µg/mL, µM	µg/mL, µM	µg/mL, µM
*S. aureus* 118 *	32, 70.1	256, 9.3	1.7, 3.7	256, 17.5	1.9, 4.2	128, 4.4	0.8, 1.8
*S. aureus* 120 *	16, 35.0	512, 18.7	3.4, 7.4	512, 35.1	3.9, 8.5	256, 8.7	1.6, 3.6
*S. aureus* 119	32, 70.1	512, 18.7	3.4, 7.4	512, 35.1	3.9, 8.5	256, 8.7	1.6, 3.6
*S. epidermidis* 127 *	16, 35.0	128, 4.7	0.9, 1.9	128, 8.8	1.0, 2.1	64, 2.2	0.4, 0.9
*S. epidermidis* 201 *	32, 70.1	256, 9.3	1.7, 3.7	256, 17.5	1.9, 4.2	128, 4.4	0.8, 1.8
*S. epidermidis* 119	16, 35.0	512, 18.7	3.4, 7.4	256, 17.5	1.9, 4.2	64, 2.2	0.4, 0.9
*E. faecalis* 110 ^#^	8, 17.5	512, 18.7	3.4, 7.4	32, 2.2	0.2, 0.5	32, 1.1	0.2, 0.5
*E. faecalis* 124 ^#^	4, 8.8	256, 9.3	1.7, 3.7	64, 4.4	0.4, 1.0	16, 0.5	0.1, 0.3
*E. faecalis* 127	8, 17.5	256, 9.3	1.7, 3.7	64, 4.4	0.4, 1.0	32, 1.1	0.2, 0.5
E. faecalis 19 ^†,#^	8, 17.5	256, 9.3	1.7, 3.7	32, 2.2	0.2, 0.5	16, 0.5	0.1, 0.3
*E. faecalis* 51 ^†,#^	4, 8.8	256, 9.3	1.7, 3.7	32, 2.2	0.2, 0.5	16, 0.5	0.1, 0.3
*E. faecium* 118 ^#^	4, 8.8	256, 9.3	1.7, 3.7	64, 4.4	0.4, 1.0	32, 1.1	0.2, 0.5
*E. faecium* 123 ^#^	2, 4.4	512, 18.7	3.4, 7.4	32, 2.2	0.2, 0.5	16, 0.5	0.1, 0.3
*E. faecium* 127	4, 8.8	512, 18.7	3.4, 7.4	64, 4.4	0.4, 1.0	32, 1.1	0.2, 0.5
*E. faecium* 3 ^†,#^	4, 8.8	512, 18.7	3.4, 7.4	32, 2.2	0.2, 0.5	16, 0.5	0.1, 0.3

^1^ By the MIC observed for G5R(66) after 24 h; ^2^ by the MIC observed for G4R(16)K(19) after 24 h; ^3^ by the MIC observed for G5R(38)K(30) after 24 h; ^4^ experiments were performed in triplicate, the concordance degree was 3/3, and ±SD was zero; * indicates methicillin resistance; ^#^ indicates vancomycin resistance; ^†^ indicates strains of marine origin isolated from seawater of the Ligurian west coast.

**Table 4 polymers-13-00521-t004:** MIC values of the three cationic empty dendrimers, expressed as µg/mL and millimolarity (mM), against the Gram-positive strains tested in the study.

	G5R(66)OH(30)MW 26,000	G4R(16)K(19)OH(13) MW 12,800	G5R(38)K(30)OH(28) MW 25,700	Commercial Antibiotics
**MIC values ^1^**	µg/mL, µM	µg/mL, µM	µg/mL, µM	µg/mL, µM ^2^
*S. aureus* 118 *	256, 9.8	256, 20.0	128, 4.98	256, 637.7
*S. aureus* 120 *	512, 19.6	512, 40.0	256, 9.96	512, 1275
*S. aureus* 119	512, 19.6	512, 40.0	256, 9.96	1, 2.5
*S. epidermidis* 127 *	128, 4.9	128, 40.0	64, 2.5	256, 637.7
*S. epidermidis* 201 *	256, 9.8	256, 20.0	128, 4.98	256, 637.7
*S. epidermidis* 119	512, 19.6	256, 20.0	64, 2.5	0.5, 1.25
				µg/mL, µM ^3^
*E. faecalis* 110 ^#^	512, 19.6	32, 2.5	32, 1.2	128, 88.3
*E. faecalis* 124 ^#^	256, 9.8	64, 10.0	16, 0.6	32, 22.1
*E. faecalis* 127	256, 9.8	64, 10.0	32, 1.2	1, 0.7
*E. faecalis* 19 ^†,#^	256, 9.8	32, 2.5	16, 0.6	32, 22.1
*E. faecalis* 51 ^†,#^	256, 9.8	32, 2.5	16, 0.6	32, 22.1
*E. faecium* 118 ^#^	256, 9.8	64, 10.0	32, 1.2	256, 176.6
*E. faecium* 123 ^#^	512, 19.6	32, 2.5	16, 0.6	128, 88.3
*E. faecium* 127	512, 19.6	64, 10.0	32, 1.2	2, 1.4
*E. faecium* 3 ^†,#^	512, 19.6	32, 2.5	16, 0.6	128, 88.3

^1^ Experiments were performed in triplicate, the concordance degree was 3/3, and ± SD was zero; * indicates methicillin resistance; ^#^ indicates vancomycin resistance; ^†^ indicates strains of marine origin isolated from seawater of the Ligurian west coast; ^2^ oxacillin; ^3^ vancomycin.

## Data Availability

Data are contained within the herein article.

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
