# Peer review of "Synthesis and Antibacterial Activity of Cationic Amino Acid-Conjugated Dendrimers Loaded with a Mixture of Two Triterpenoid Acids"

_polymers, 2021, doi:10.3390/polym13040521_

Round 1

Reviewer 1 Report

The authors describe the antibacterial activity of 3 cationic amino acid conjugated dendrimers loaded with a mixture of two triterpenoid acids. The specific field of study – the development of therapeutic alternatives to antibiotic resistant species - is of high general interest. However, there are a number of important issues which must be addressed before the paper can be considered for publication

While a lot of data in invoked, very little new experimental little data is actually presented, with the notable exception of the data Table 2. In fact, even the beginning of the results section contains a lot of considerations which are of a general nature rather than experimental results. This absence of the experimental data makes it difficult to evaluate the manuscript from the points-of-view of both the suitability of methods used and the overall scientific importance of the results. 

Of particular concern is the data relating to the synthesis of the dendrimers. The authors state that the dendrimers “prepared in this study are original antibacterial cationic devices, because in place of using commercially available highly cationic and not biodegradable PAMAM scaffolds, as host macromolecules for the transport and deliver of UOA, they were totally synthetized in our laboratory and harmonized hydrolysable ester-based uncharged matrices, with cationic shells made of natural amino acids.” Furthermore, they state that “Robust and optimized synthetic procedures were performed to prepare the UOA-loaded polyester-based amino acids-conjugated cationic dendrimers as described in details in Section 3.”

I cannot find any detailed description of synthetic procedure in section 3 (or section 2). There also does not appear to be any supporting information. Some references are given to methods used in previous papers, but this lack of experimental data makes it very difficult to appreciate various, in my view, important issues. For example:

  1. How is the exact number of conjugated amino acids controlled in the synthesis and how is it the number in the final system checked?
  2. How much of the surface of the dendrimers is covered with attached amino acids?
  3. Particle size and particle size distribution are important for pharmaceutical applications. Particle sizes are simply quoted.
  4. Can the UOA loading be controlled or are the loadings simply the result of an identical synthesis procedure for the 3 dendrimers?
  5. What specific evidence is there that the acids are in fact encapsulated in the dendrimers, rather than just physically mixed? How are the CH2OH groups of the dendrimers diagnostic to confirm the encapsulation of the UAO (L 281) ? Are these signals different in the unloaded dendrimers? Some indication of how UAO might be released would also be of interest.
  6. In Figure 2. Do the subscripts for the OH indicate the exact number of these groups? This could be specified in the legend. How is this measured or calculated?
  7. The authors mention “estimating” the MW of the loaded dendrimers (L 296-297). How was it estimated? Could it be somehow measured?

With regard to Table 2. The authors conclusions on the seem to be fully supported by the data, however, the situation is a lot less clear for S. epidermis and S. aureas. For example, when comparing G5R(66) and G4R(16)K(19), it is difficult to see how the activity of G5R(66) is higher than that of G4R(16)K(19). Only the MIC values for S. epidermis 219 is different. The uncertainty should be specifically addressed.

The antimicrobial activity of the unloaded dendrimers should be tested for comparison with the values in Table 2. As indicated by the authors, it seems certain that the activity observed is largely due to the dendrimers alone. This should be confirmed. Comparison with a commercial antibiotic could also be useful.

The origin of all the data in Table 1 is not clear. In particular, how the number of moles of UOA per dendrimer and the amount of UOA released after 24 h were determined

It is difficult to see what the NMR of the physical mixture of UA and OA adds to the discussion (Fig 3). A 1:1 mixture can only realistically be that and the spectrum would be expected to be a mixture of the two individual spectra. On the other hand, showing the spectra of all 3 devices could however be useful instead of just one as an example. In any case, I believe that all the NMR signals should be fully assigned in the figures.

Minor issues

The reason for the use of a mixture of ursolic acid (UA) and oleanolic acid (OA) should be specified and, in necessary, referenced in the introduction.

The authors should define all acronyms used in the in the text, for example, MDR, PMAMs, CFU

L 75: In this background; against this bagground?

L106: what was the purity observed by GC??

L 97: close to the OH groups? Should this be close to amino acid groups?

L379: were not effective

Author Response

Comments and Suggestions for Authors

The authors describe the antibacterial activity of 3 cationic amino acid conjugated dendrimers loaded with a mixture of two triterpenoid acids. The specific field of study – the development of therapeutic alternatives to antibiotic resistant species - is of high general interest. However, there are a number of important issues which must be addressed before the paper can be considered for publication

While a lot of data in invoked, very little new experimental little data is actually presented, with the notable exception of the data Table 2. In fact, even the beginning of the results section contains a lot of considerations which are of a general nature rather than experimental results. This absence of the experimental data makes it difficult to evaluate the manuscript from the points-of-view of both the suitability of methods used and the overall scientific importance of the results.

Of particular concern is the data relating to the synthesis of the dendrimers. The authors state that the dendrimers “prepared in this study are original antibacterial cationic devices, because in place of using commercially available highly cationic and not biodegradable PAMAM scaffolds, as host macromolecules for the transport and deliver of UOA, they were totally synthetized in our laboratory and harmonized hydrolysable ester-based uncharged matrices, with cationic shells made of natural amino acids.” Furthermore, they state that “Robust and optimized synthetic procedures were performed to prepare the UOA-loaded polyester-based amino acids-conjugated cationic dendrimers as described in details in Section 3.”

I cannot find any detailed description of synthetic procedure in section 3 (or section 2).

The authors apologize for their distraction and thank the Reviewer for pointing out that they incorrectly refer to Section 3 instead of Section 2. Section 3 has been replaced with Section 2 (page 11, line 358).

There also does not appear to be any supporting information. Some references are given to methods used in previous papers, but this lack of experimental data makes it very difficult to appreciate various, in my view, important issues.

We agree with the Reviewer and thank him for his suggestion. Although the preparation and characterization of UOA-loaded dendrimers was already present in the Section 2.2, of the unrevised manuscript, an additional experimental part concerning the synthesis and characterization of cationic empty dendrimers is useful, so that readers and the Reviewers can easily consult it and speculate on synthetic and characterization procedures. Therefore, the requested description of the synthesis and characterization of empty dendrimers has been added in Section 2.2., before the procedure employed to prepare the physical mixture of UA and OA (pages 4-5, lines 132-176), and the relative results and discussion in Section 3.2 (pages 10-11, lines 298-340). Other parts of the original manuscript have been modified, accordingly. A Supporting Information (SI) file has been drawn up, in which significant images, relative to this new part, have been inserted.

For example:

How is the exact number of conjugated amino acids controlled in the synthesis and how is it the number in the final system checked?

Following the Reviewer’s suggestion and the insertion of the experimental part relating to the preparation of empty cationic dendrimers, as well as the insertion of the related discussion, the answer to the Reviewer's question is now present in the text (pages 10-11, lines 302-320). However, the exact number of conjugated amino acids was controlled by NMR analysis, as well as their number in the final system. As example, an image (Figure S3) has been included in the new SI.

How much of the surface of the dendrimers is covered with attached amino acids?

Also for this point, the explanation is present in the new insertions in Sections 2.2 and 3.2. Depending on the dendrimer considered, the surface of the dendrimers was more or less covered by with amino acids. In particular, the G4 dendrimer had the surface covered by 73%, while the G5 dendrimer, containing only Arginine, by 69% and the other G5 dendrimer by 71%. However, these data were also included in Table 2 (revised manuscript).

Particle size and particle size distribution are important for pharmaceutical applications. Particle sizes are simply quoted.

Regarding this Reviewer’s request, we apologize but we disagree with him, because particle sizes were not simply mentioned. On the contrary, being well aware of the importance of this data for a device for biomedical applications, a discussion on the particle size of our devices, and also on the value of Z potential, had already been included in the unrevised form of our manuscript (page 14, lines 421-437). We report this dissertation below.

Since devices aiming at a future clinical application have to possess peculiar requisites, in terms of water-solubility, particles dimensions and surface charge, the determination of their values and a brief discussion of the results obtained is herein mandatory. The particle sizes of UOACDs ranged from 16 to 25 nm, while Z-potentials were positive and ranged from 25 to 34 mV. It is known that small particles assure minor tissue toxicity, but extremely minute particles could easily undergo hepatobiliary and renal clearance [36]. A correct balance to minimize both tissue toxicity and fast clearance by the mononuclear phagocytic system (MPS) is the best solution and, as reported, particles less than 100 nm and greater than 20 nm could be a good choice [38,39].

The average particle size of UOACDs was in any case less than 100 nm, thus assuring that during a possible in vivo administration, they do not generate administration embolism, thus also being suitable for intravenous or intraperitoneal administration to the patient. Unfortunately, G5R(66)UOA(3) particles displayed a mean size < 20 nm, and short circulation time could affect its efficiency in vivo. The surface charge of UOACDs had higher values for G5 samples, and increased at the increase of the number of cationic groups. The UOACDs displayed high water-solubility, providing solutions stable along time, as announced by Z-potential values around 30 mV.

Can the UOA loading be controlled or are the loadings simply the result of an identical synthesis procedure for the 3 dendrimers?

UOA loading has been controlled by the mean of 1H NMR analysis. In order to better answer to the Reviewer’s request, we have cited below what we have reported in the text. Part of what reported was already present in the original manuscript, while a new part has been inserted for greater clarity.

The peaks under 1.0 ppm, not present in the parent dendrimer spectrum, and belonging to the UOA mixture, are relative to the seven CH3 groups and H (C(5)) of UA and UO, for a total of 22 H [37], while the broad peak at 3.7 ppm refers to the methylene in the CH2OH groups of dendrimers. These peaks were very diagnostic to confirm the success of the encapsulation reaction, and the integral value associated to these peaks, was used to estimate the number of UOA units encapsulated per dendrimer mole. Briefly, the number of UOA units per complex mole was obtained comparing the integral value of CH2OH of the dendrimer scaffolds at 3.6–3.8 ppm, which was associated to 13, 60 and 56 proton atoms for G4R(16)K(19)UOA(4), G5R(66)UOA(3) and G5R(38)K(30)UOA(8) respectively and the value of the integral of peaks between 0.72 and 0.98 ppm, which was associated to 22 H.

What specific evidence is there that the acids are in fact encapsulated in the dendrimers, rather than just physically mixed?

As reported in the experimental part (Section 2.4):

“…after the removal of the solvent at reduced pressure, the obtained white solids were suspended in dichloromethane (DCM) overnight to wash away the free UOA not entrapped. The solids not dissolved were decanted, the DCM was separated and evaporated to obtain a white solid identified as UOA by IR analysis.”

However, an additional new part which clarify this point has been included in the discussion section 3.4 (page 12, lines 365-370).

How are the CH2OH groups of the dendrimers diagnostic to confirm the encapsulation of the UAO (L 281) ?

Actually, we did not say that the CH2OH signal was diagnostic to confirm the encapsulation of the UAO, but that the coexistence of the signals below 1.00 ppm and of the CH2OH signal in the spectra of the complexes, representing the presence of UOA and of the parent dendrimer respectively, were diagnostic of the encapsulation. Furthermore, we have said that by comparing their integrals it was also possible to have evidence of how much UOA had been loaded. However, thinking that if this doubt came to the Reviewer it could also come to the readers, clarification sentences have been inserted in the discussion (page 13, among lines 395-406) and an explicative Figure (Figure S57 has been included in the SI.

Are these signals different in the unloaded dendrimers?

We are confident that the answer to this point has been already included in the answer to the previous point. Anyway, concerning the signals of CH2OH, as observable also in Figure S7 and described in its caption, in the unloaded dendrimers they appears slightly shifted due to the different solvents used during spectra acquisition.

 In Figure 2. Do the subscripts for the OH indicate the exact number of these groups? This could be specified in the legend. How is this measured or calculated?

At the suggestion of the Reviewer, the following sentence is now included in the text (lines 103-105):

“Note that, the final dendrimers retained a variable number of free hydroxyls during functionalization, which has been indicated with the numbers subscripted close to the OH groups in Figure 2 and which contributed to their water solubility.”

Rationally, the number of OH groups in the UOACDs must be the same of that of the empty dendrimers and that number was determined by 1H NMR analysis as already explained.

The authors mention “estimating” the MW of the loaded dendrimers (L 296-297). How was it estimated?

The MW was estimated simply by summing the MW of the forerunner dendrimer to the MW of UOA multiplied by the number of complexed units as deduced by NMR spectra. A sentence similar to this one has been inserted in the text for more clarity (page 13, lines 417-420).

Could it be somehow measured?

Off course. Among the most used techniques we find the MALDITOF but being a very expensive technique, not all laboratories have it, including ours.

With regard to Table 2. The authors conclusions on the seem to be fully supported by the data, however, the situation is a lot less clear for S. epidermis and S. aureas. For example, when comparing G5R(66) and G4R(16)K(19), it is difficult to see how the activity of G5R(66) is higher than that of G4R(16)K(19). Only the MIC values for S. epidermis 219 is different. The uncertainty should be specifically addressed.

We apologize to the Reviewer for not agreeing with his comment, but regarding the MIC values of the two dendrimers  G5R(66) and G4R(16)K(19), shown in Table 2 (now Table 3) against Staphylococci, with the exception of S. epidermidis 119, all MICs reported for G5R(66) are significantly lower than those of G4R(16)K(19), and consequently G5R(66)  can be defined as generally more active. Please, see at the MICs reported:

G5R(66)UOA(3)

MW 27400

G4R(16)K(19)UOA(4) MW14600

MIC values

 µM

µM

S .aureus 118*

 9.3

17.5

S. aureus 120*

18.7

35.1

S. aureus 119

18.7

35.1

S. epidermidis 127*

4.7

8.8

S. epidermidis 201*

9.3

17.5

S. epidermidis 119

18.7

17.5

 However, for more clarity, the exception of S. epidermidis 119 has been highlighted in text of the revised manuscript (page 17, line 510).

The antimicrobial activity of the unloaded dendrimers should be tested for comparison with the values in Table 2. As indicated by the authors, it seems certain that the activity observed is largely due to the dendrimers alone. This should be confirmed. Comparison with a commercial antibiotic could also be useful.

As suggested by the Reviewer, the antimicrobial activity (MICs) of the unloaded dendrimers has been tested for allowing a comparison with the MIC values in Table 2 (now Table 3), and the results were reported in the new Table 4. Also commercial antibiotics were included, for comparison. A discussion of the results, which confirmed our speculations, has also been added (page 15, lines 478-494).

The origin of all the data in Table 1 is not clear. In particular, how the number of moles of UOA per dendrimer and the amount of UOA released after 24 h were determined.

We have already explained how the number of moles of UOA per dendrimer were determined in a previous point and in the text at page 13, in the sentences after Figure 3. Concerning the release of UOA, it was explained by inserting a new part in the discussion section at page 14 (lines 438-451) and a Figure S8 in SI.

It is difficult to see what the NMR of the physical mixture of UA and OA adds to the discussion (Fig 3). A 1:1 mixture can only realistically be that and the spectrum would be expected to be a mixture of the two individual spectra. On the other hand, showing the spectra of all 3 devices could however be useful instead of just one as an example. In any case, I believe that all the NMR signals should be fully assigned in the figures.

As suggested by the Reviewer, the original Figure 3, has been removed. The spectra of all 3 devices have been included in the work, one in the main text (Figure 3) and two in the SI (Figure S5 and S6). Where possible, the NMR signals have been assigned.

Minor issues

The reason for the use of a mixture of ursolic acid (UA) and oleanolic acid (OA) should be specified and, in necessary, referenced in the introduction.

As requested, the reason for the use of a mixture of ursolic acid (UA) and oleanolic acid (OA) has been specified in the introduction, by adding the following sentence and the relative reference (page 3, lines 85-88).

“We decide to use a mixture of UO and OA, in place of pure compounds because when extracted from medicinal plants, they were achieved as a mixture and because both of them are endowed with similar beneficial properties, including the antibacterial one [26].”

The authors should define all acronyms used in the in the text, for example, MDR, PMAMs, CFU

It was done.

L 75: In this background; against this bagground?

The requested change has been applied..

L106: what was the purity observed by GC??

It means that the purity was assessed by the manufacturer by GC analysis.

L 97: close to the OH groups? Should this be close to amino acid groups?

The sentence has been reformulated for more clarity.

L379: were not effective.

The suggestion was applied.

Reviewer 2 Report

This manuscript presents the obtaining of  three polyester based-amino-acids-conjugated dendrimers, loaded with a 1:1 physical mixture of commercially available ursolic acid (UA) and oleanolic acid (OA) (UOA), and the evaluation of their antibacterial effect against 15 Gram-positive clinical isolates, with excellent results.

The experiments in this study are well planned and of good technical quality. However, several issues need to be addressed for the manuscript to be published in Polymers. To improve the manuscript I suggest the following comments:

  • the complexes obtained are called devices, I would suggest replacing it with compounds, or agents as it is also used in the text
  • the presentation of the complexes and their characteristics (line 140-162) can be structured in a table, so the differences between them can be traced more easily;
  • antimicrobial tests were determined only by determining the minimum inhibitory concentrations and the standard deviation was 0. It would be useful to do for verification other methods such as diffusion.

Author Response

This manuscript presents the obtaining of three polyester based-amino-acids-conjugated dendrimers, loaded with a 1:1 physical mixture of commercially available ursolic acid (UA) and oleanolic acid (OA) (UOA), and the evaluation of their antibacterial effect against 15 Gram-positive clinical isolates, with excellent results.

The experiments in this study are well planned and of good technical quality. However, several issues need to be addressed for the manuscript to be published in Polymers. To improve the manuscript I suggest the following comments:

the complexes obtained are called devices, I would suggest replacing it with compounds, or agents as it is also used in the text

As suggested by the Reviewer, devices has been replaced by compounds or agents throughout the manuscript.

the presentation of the complexes and their characteristics (line 140-162) can be structured in a table, so the differences between them can be traced more easily;

As suggested by the Reviewer, the characteristics of the complexes have been structured in a Table (namely Table 1). The numbers of the subsequent Tables have been updated accordingly.

antimicrobial tests were determined only by determining the minimum inhibitory concentrations and the standard deviation was 0. It would be useful to do for verification other methods such as diffusion.

As the Reviewer probably knows, the agar diffusion test is a qualitative method that is certainly less precise than the determination of the MIC values, which is instead a quantitative test, and which, for this reason, is always preferable used.
The agar diffusion test may possibly precede the study of the MICs, especially in the case in which the activity of new molecules is studied, but the determination of the MIC values, in the final analysis, is always preferable since, referring to a concentration, provides much more accurate information.

References

CLSI (NCCLS), 2003a. Methods for Dilution Antimicrobial Susceptibility Tests forBacteria that Grow Aerobically: Approved Standard 23. National Committee forClinical Laboratory Standards, Wayne, PA, USA.

CLSI (NCCLS), 2003b. MIC Testing Supplemental Tables: Approved Standard M100.National Committee for Clinical Laboratory Standards, Wayne, PA, USA.

CLSI (NCCLS), 2003c. Methods for Dilution Antimicrobial Susceptibility Tests forBacteria that Grow Aerobically; Approved Standard M7-A6. National Committee forClinical Laboratory Standards, Wayne, Pa, USA

Round 2

Reviewer 1 Report

With regard to the MIC Values in the new Table 3, I accept that I misread some of the data for my original data and I would even suggest that the authors present this data as they did in their response to me.

In my original report my main concern was the lack of data, which, for me, made an evaluation of the manuscript problematic. The example list of questions was related to this lack of detail. The main problem was not that section 3 was written instead of section 2, but that section 2 alone did not have enough detail.

The authors have misconstrued a number of my questions. For example, I am aware that MS can be used to determine the molecular weight (I do have access to the technique) and my question was whether the authors could include this data, if possible naturally. I would not insist on this data. Similarly, when I asked if the loading could be controlled, my meaning was could a different number of UAOs be inserted during the syhthesis, not can the number inserted be checked afterwards by NMR. For the particle sizes, I simply wanted to know how the particle sizes and the size distribution were measured. I did see that they these data were discussed, with the relevant sizes mentioned (or cited). This applies to any of the measured values, such as the number of free OH groups of the surfaces of the dendrimers. For the purity, I thought the percentage purity cited by the manufactured might be available.

As an example of my misgivings, I do not think that integration of NMR peaks necessarily indicates that encapsulation has taken place, or that washing the solid necessarily removes all of the “external” UAO, however, shifts in NMR signals (for example the CH2OH signals) with respect to free UAO,  could provide very convincing evidence of encapsulation. This happens in other situations with which I am more familiar. In my view it is important that the reader to have enough experimental evidence to evaluate the results presented and decide on the hypothesis proposed by the authors.

In the revised version, the authors have included a lot of new information in both the main manuscript and supporting information which do somewhat address my concerns. I now believe that additional information which is necessary is now available and that the manuscript can be published.

Reviewer 2 Report

The manuscript has been improved according to the reviewers' suggestions and therefore I agree with the acceptance of the present form.